# Circulating Biomarkers Involved in the Development of and Progression to Chronic Pancreatitis—A Literature Review

**DOI:** 10.3390/biom14020239

**Published:** 2024-02-18

**Authors:** Valborg Vang Poulsen, Amer Hadi, Mikkel Parsberg Werge, John Gásdal Karstensen, Srdan Novovic

**Affiliations:** 1Pancreatitis Center East, Gastrounit, Copenhagen University Hospital—Amager and Hvidovre, 2000 Copenhagen, Denmark; valborg.vang.poulsen.02@regionh.dk (V.V.P.); amer.hadi@regionh.dk (A.H.); mikkel.parsberg.werge@regionh.dk (M.P.W.); john.gasdal.karstensen@regionh.dk (J.G.K.); 2Department of Clinical Medicine, University of Copenhagen, 2000 Copenhagen, Denmark

**Keywords:** acute pancreatitis, chronic pancreatitis, fibrosis, inflammation, oxidative stress

## Abstract

Chronic pancreatitis (CP) is the end-stage of continuous inflammation and fibrosis in the pancreas evolving from acute- to recurrent acute-, early, and, finally, end-stage CP. Currently, prevention is the only way to reduce disease burden. In this setting, early detection is of great importance. Due to the anatomy and risks associated with direct sampling from pancreatic tissue, most of our information on the human pancreas arises from circulating biomarkers thought to be involved in pancreatic pathophysiology or injury. The present review provides the status of circulating biomarkers involved in the development of and progression to CP.

## 1. Background

Acute pancreatitis (AP), chronic pancreatitis (CP), and pancreatic ductal adenocarcinoma (PDAC) place a significant burden on healthcare systems worldwide. AP is among the three most common benign gastrointestinal diseases, with a mortality rate of 0.9% and an estimated economic burden of USD2.6 billion per year in the US [1]. CP is characterized by gradual irreversible damage to the endocrine and exocrine parenchyma caused by inflammation and subsequent replacement of these tissues with fibrotic tissue and atrophy [2]. Over the last two decades, the incidence of CP has increased by 50%, and there are currently no treatments available to alter this disease’s course, resulting in significantly reduced life expectancy and quality of life. Prevention is the only way to reduce the disease burden, as serious complications including exocrine pancreatic insufficiency, malabsorption, diabetes mellitus, and PDAC may evolve as this disease progresses [3]. 

Approximately 50% of patients with CP have a history of AP [3]. There is continual replacement of the pancreatic tissue with fibrosis. Individuals who experience first-time AP have a 22% chance of developing recurrent acute pancreatitis (RAP) [4], and patients who experience three episodes of RAP have a 16% chance of developing CP. In addition, patients with four or more episodes of RAP have a much higher risk, around 50%, of developing CP [5]. 

Thus, the continuum from the first episode of AP to the manifestation of CP provides a framework for epidemiologic studies and the time-dependent evolution of circulating biomarkers involved in the progression of this disease. 

This review aims to compile and synthesize findings on existing human studies on biomarkers thought to be involved in the development and progression of CP. 

## 2. Materials and Methods

The literature search was conducted using PubMed. The search was performed to a cut-off date of 1 September 2023 to ensure the inclusion of the most relevant and up-to-date studies. The main search included a combination of text words and MeSH terms: (inflammation, oxidative stress, and fibrosis), combined with chronic pancreatitis and (serum, plasma, or biomarker). Additionally, cross-references were identified manually through the citation list of selected studies to capture additional sources. To ensure completeness, a search was conducted on PubMed for each biomarker mentioned in the review. 

We included human studies that compared blood/serum/plasma biomarkers of inflammation, fibrosis, and oxidative stress in patients with CP compared to healthy controls. Animal studies, studies evaluating tissue biopsy biomarkers, and studies evaluating cancer biomarkers were excluded. All the figures included were generated using BioRender.com.

## 3. Results

We identified 96 studies spanning from 1981 to 2022, examining 126 different biomarkers, with 59 being examined multiple times. Table 1, Table 2 and Table 3 provide an overview of the biomarkers examined at least twice, with brief additional information, their potential role in pancreatitis, and their levels in patients with CP. Table 4 provides an overview of additional biomarkers only examined once. 

### 3.1. Inflammation

From 1994 to 2022, 55 articles examined 64 inflammatory biomarkers, of which 23 were examined multiple times. In addition, five of these studies included patients with AP [6,7,8,9,10]. The key findings are summarized in Table 1. Of the 23 biomarkers, 12 were either not elevated in CP or the findings were inconclusive. Several pro-inflammatory interleukins (IL-6, IL-8, and IL-12) were found to be elevated in patients with CP compared to healthy controls [11,12,13], along with vascular endothelial growth factor (VEGF), intercellular adhesion molecule (ICAM), chemerin, fractalkine, resistin, osteopontin, and neopterin [7,10,14,15,16,17,18]. In contrast, leptin was found to be reduced in patients with CP [16]. IL-1β, IL-6, IL-10, tumor necrosis factor α (TNF-α), adiponectin, and leptin were the most studied inflammatory biomarkers in CP. However, the findings were cohesive only for IL-6, TNF-α, and leptin. IL-10, IL-12, TNF-α, and INF-γ were elevated in patients with AP [6,8]. Figure 1 demonstrates a schematic overview of the inflammatory biomarkers involved in the progression to CP.

#### 3.1.1. Interleukin 6 

IL-6 induces the synthesis of acute-phase proteins and the production of other cytokines, including C-reactive protein (CRP) [19]. Sixteen studies measured IL-6, with twelve observing higher levels in patients with CP compared to healthy controls [6,11,12,20,21,22,23,24,25,26,27,28], although the difference was not significant in three studies [20,25,28]. Four studies found no difference in the IL-6 levels [27,29,30,31]. In one study, a surge in IL-6 serum levels was observed in patients with alcoholic CP after the consumption of alcohol, with a decrease to the pre-stimulatory levels after 4–24 h, suggesting a correlation between alcohol consumption and IL-6 levels [24]. Elevated IL-6 levels were also evident in AP [6]. IL-6 rises 1–2 days before CRP, making it suitable for an early distinction between severe and mild AP [32,33]. Higher concentrations of IL-6 are linked to the increased risk of complications and death in severe AP [34,35,36,37]

#### 3.1.2. Tumor Necrosis Factor α

TNF-α is a cytokine that facilitates both inflammation and fibrosis formation. It plays a key role in regulating other cytokines towards inflammation and activating pancreatic stellate cells (PSCs). TNF-α triggers the activation of PCSs, which, in turn, start producing extracellular matrix (ECM). This disorganization of the ECM leads to fibrosis formation and chronic inflammation of the pancreas [38,39]. The levels of TNF-α in patients with CP were investigated in 12 studies from 1999 to 2022. Elevated levels were found in six studies [9,11,22,31,39,40,41], one found lower levels [9], while the remaining five found no significant differences [8,20,25,29,42]. Kiyci et al. discovered significantly higher serum levels of TNF-α in AP compared to CP, indicating TNF-α’s potential role in the progression of the disease. However, it is worth noting that this study included only 13 patients with AP, 36 patients with CP, and 14 controls [8]. 

#### 3.1.3. Leptin

Leptin, an adipokine with a crucial role in metabolism, obesity, and cardiovascular diseases, has also been found to activate macrophages and T-lymphocytes, stimulating their cytokine secretion [43]. Moreover, it has been demonstrated to induce fibrosis in the liver by inhibiting hepatic stellate cell apoptosis [16,44,45]. Five studies found reduced levels of leptin in patients with CP [16,46,47,48,49], while one study found elevated leptin levels compared to healthy controls [41]. Because leptin is secreted by adipocytes, a higher fat percentage results in a higher amount of circulating leptin. Patients with CP had lower BMI across the involved studies, making it difficult to determine if the reduced levels were due to pancreatitis or to a lower fat mass. Additionally, patients with CP with diabetes mellitus (DM) were found to have higher levels of leptin than patients with CP without DM [41]. Lower levels of leptin may play a protective role in the development of CP by increasing apoptosis of the PSCs.
biomolecules-14-00239-t001_Table 1Table 1An overview of the inflammatory biomarkers involved in the development and progression of chronic pancreatitis. (Symbols in the column “Blood Levels in Patients with CP” correspond to the reference in the column “Biomarkers”; =: no difference; ↓: reduced levels; and ↑: elevated levels).BiomarkersMechanismPancreas-Specific EffectsBlood Levels in Patients with CP CommentIL-1β [12,22,29,30,31,50]A pro-inflammatory cytokine that activates several intracellular responses, e.g., stimulation of IL-6, IL-8, and TNF-α [51,52].Excessive or prolonged IL-1β activation can lead to CP [52,53]. Activates proliferation and collagen secretion in fibroblasts [54].= = ↓ ↓ ↑ =Overexpression of IL-1β in murine pancreas results in CP [53]. Increases protease inhibitors having a protective effect in CP [30].IL-1α[9,29,50]A pro-inflammatory cytokine that induces inflammation via activation of, e.g., COX2, IL-6, and TNF-α [55].Not typically associated with the pancreas but can indirectly be involved in pancreatic diseases.↓ ↓ =
IL-1Ra[12,29]An IL-1 receptor antagonist. Anti-inflammatory cytokine with protumor activity [12].Has a protective effect on both AP and CP [56,57].↑ =Higher levels in PDAC compared to CP [29].IL-2[6,29,58]A potent Th1-related cytokine that acts on NK cells and T-cells [59].Increases T-cells in the pancreas and induces expression of T-cell-associated proteins [60].↑ ↓ ↓
IL-2R[20,61,62,63]IL-2 receptor↑ = = =
IL-4[6,9,29]Modulates the differentiation of precursor Th cells to Th2 cells [64]; inhibition of pro-inflammatory cytokine synthesis [65].Secreted by PSCs, mediates macrophage activation by participating in the promotion of pancreatic fibrosis [66].↓ ↑ ↓Potentially, levels of IL-4 in patients with CP depend on whether inflammation or fibrosis is the dominant process.IL-6[6,11,12,20,21,22,23,24,25,26,27,28,29,30,31,67]A pro-inflammatory cytokine that causes cell proliferation, differentiation, and inflammatory responses and triggers the synthesis of acute-phase proteins [19,30].Promotes PSCs activation and collagen synthesis through the upregulation of TGF-β1 [68].↑ ↑ ↑ = ↑ ↑ ↑ ↑ = ↑ = = = = = ↑Levels are closely linked to the quantity of alcohol consumed by patients with alcoholic CP [24]. Elevated in AP and reflects the severity and prognosis of the pancreatitis [37].IL-8[9,11,12,31]A chemoattractant that acts as a neutrophil activator and a pro-angiogenic factor [9,69].Circulating neutrophils from patients with CP express mRNA for IL-8 [69]. High levels of IL-8 are found in CP tissue [69,70,71].↑ ↑ ↑ ↑Depending on the etiology, the amount of IL-8 correlates with the severity of the pancreatitis [69,70,71].IL-10 [6,23,29,31,58]An anti-inflammatory cytokine that inhibits cytokine release from lymphocytes, e.g., IL-12, IL-6, and TNF-α [13,72].Has a protective effect on the pancreas during inflammation. The absence of IL-10 prevents the downregulation of inflammation [72,73].↓ ↑ ↑ = =Is seen to have a protective effect on the pancreas in mice [74].IL-12[6,9,13,29]Activates Th1-cells and induces the secretion of cytokines, e.g., INF-γ, IL-2, and TNF-α [6].The level escalates during the transition from AP to CP. Increased levels in both conditions [6].↑ ↑ ↑ ↓Potential role in the progression of the disease [6].IL-17[23,75]A pro-inflammatory cytokine with a key role in the initial immune response [76].Triggers damage to pancreatic acinar cells by producing and releasing cytokines/chemokines recruiting immune cells [76].= ↑Valuable severity and prognostic factor in AP progression [77].GM-CSF[46,78]A growth and differentiation factor for granulocytes and macrophages [79].Regulates cancer-associated inflammation in PDAC [80].↓ ↑
IFN-γ[6,13,29]A pro-inflammatory cytokine produced by activated T-cells and NK-cells, with chemotactic abilities [81].Stimulated by upregulated IL-18 and IL-12 in CP [6]. Elevated levels were found in CP tissue [82,83,84].↑ = ↓Potential role in the progression of pancreatitis [6].TNF-α[8,9,11,20,22,25,29,31,39,40,41,42]Regulates cytokines and adhesion molecules; also, a priming activator of inflammatory cells and PSCs [38].Induces PSC activation and collagen synthesis leading to fibrosis and inflammation in the pancreas [40].= ↓ ↑ = ↑ = = ↑ ↑ ↑ ↑ =Elevated levels are also seen in patients with AP [33,36].ICAM[15,85,86,87]An adhesion molecule that serves to mediate the adhesion of immune cells to endo-/epithelial cells [88].Overexpression of ICAM-1 in pancreatic endothelial cells leads to inflammatory cell infiltration in the pancreatic parenchyma [88].↑ = ↑ ↑Elevated levels in AP correlate with higher mortality rates and necrosis development [88]VEGF[14,89]A pro-angiogenic mediator that enhances vascular permeability and stimulates immune cell migration [90].Not typically associated with the pancreas, it can indirectly be involved in pancreas diseases.↑ =
Fractalkine [7,91,92]Adhesion molecule that can be cleaved and functions as a chemoattractant [93].Expressed on the cell membranes of PSCs, it induces monocyte recruitment in the inflamed pancreas [7].↑ ↑ ↑Alcohol consumption influences the levels of fractalkine. One study only found elevation in mild and severe CP [91].Chemerin[10,94,95]An adipokine with chemoattractant properties, promotes the differentiation of adipocytes [96].Promotes the recruitment of macrophages to the inflamed pancreas [96].↑ ↑ ↑No correlation between chemerin levels and alcohol intake or diabetes [94].Adiponectin [16,22,41,47,48,97]An adipokine with anti-inflammatory properties. Reduces the levels of circulating fatty acids, activates their oxidation, and prevents lipid accumulation in cells [98,99].A lack of adiponectin accelerates the progression of CP in mice [100].= ↑ ↓ = = ↑Levels are inversely proportional to fat percentage.Leptin[16,41,46,47,48,49]An adipokine with pro-inflammatory and pro-fibrogenic properties [44,45].Inhibits SC apoptosis; therefore, lower levels are thought to induce SC apoptosis and thereby inhibit fibrosis [16].↓ ↑ ↓ ↓ ↓ ↓Higher levels were found among patients with CP with DM [41].Resistin[16,46,101]An adipokine that acts in a pro-inflammatory manner by upregulating IL-6 and TNF-α [102].Increases the concentration of TNF-α, which, in turn, activates PSCs [16].↑ ↑ ↑Higher levels were found among patients with CP with DM [41].Osteopontin [17,103,104,105,106]A glycophosphoprotein produced and secreted by osteoblasts, activated T cells, macrophages, and others. Functions as a chemoattractant in sites of inflammation [17,107].May play a part in the calcification and the formation of pancreas calculi [108].↑ ↑ ↑ = =
Neopterin[18,20,62]A compound secreted by activated macrophages stimulated by INF-γ [20].A marker of the cellular immunity mediated by the lymphocyte–macrophage axis [20].= ↑ ↓Elevated in patients with AP and can reflect the severity and prognosis of AP [109,110].DM: diabetes mellitus; GM-CSF: granulocyte-macrophage colony-stimulating factor; ICAM: intracellular adhesion molecule; IFN-γ: interferon-γ; NK: natural killer; PDAC: pancreatic ductal adenocarcinoma; and VGEF: vascular endothelial growth factor.


### 3.2. Fibrosis

A total of 46 studies spanning from 1995 to 2022 examined 28 potential biomarkers of fibrosis in patients with CP. Of these, 13 were studied multiple times. Three studies also included patients with AP [7,10,111]. Table 2 provides an overview of the examined biomarkers. The biomarkers mainly consist of PSCs activators, with the most extensively studied being TGF-β, PDGF, and MIC-1, and components of the ECM, with TIMP-1 and MMP-9 being studied the most. Figure 2 demonstrates a schematic overview of the fibrotic biomarkers associated with the development of CP.

#### 3.2.1. Extracellular Matrix Remodeling

Continuous modulation of the ECM leads to fibrosis. Matrix metalloproteinases (MMPs) degrade the ECM, while tissue inhibitors of matrix metalloproteinases (TIMPs) inhibit MMPs. Numerous studies have measured the concentration of these biomarkers in patients with CP. MMPs are included in seven of the studies we reviewed [39,47,48,87,112,113,114]. Elevated levels of MMP-1, MMP-2, MMP-7, and MMP-9 were found in patients with CP compared to the control group, while MMP-3 was not seen to be elevated in patients with CP. 

TIMP-1 concentrations in patients with CP were studied in nine of the studies we reviewed [15,48,85,87,105,106,113,115,116], all showing elevated concentrations in patients with CP, although three did not reach significance [48,85,105]. Hyaluronic acid (HA), laminin, and fibronectin are also important components of the ECM and are directly associated with the potential role of the ECM in the context of CP, see Figure 2. Elevated levels of all these components of the ECM were found in patients with CP. Four studies found elevated levels of HA [92,101,117,118], a fundamental component of the ECM in the pancreas. The Mac-2-binding protein (M2BP), a ligand which binds to ECM proteins and a novel biomarker of liver fibrosis, has also been found to be elevated in patients with CP. Additional biomarkers of the ECM and their potential role in CP are shown in Table 4. 

#### 3.2.2. Activation of PSCs

The activation and proliferation of PSCs influence the development of pancreatic fibrosis by the synthesis and remodeling of the ECM. The remodeling of the ECM is primarily mediated through the PCSs’ secretion of MMP and TIMP [119]. 

The cytokine transforming growth factor β1 (TGF-β1), the growth factor platelet-derived growth factor (PDGF), and the chemokine monocyte chemoattractant protein 1 (MCP-1) are among the most important mediators involved in the activation of PSCs. With few exceptions, these biomarkers are all found to be elevated in patients with CP compared to the controls. Elevated levels of TGF-β, PDGF, and MCP-1 were also found in patients with AP [7,10,111]. Macrophage inhibitory cytokine 1 (MIC-1), a cytokine part of the TGF-β family, has also been found to be elevated in patients with CP in five different studies. Its specific role in the pancreas is not extensively studied, but, as a part of the TGF-β family, it can be presumed that it has a role in the activation of PSCs. 

Some of the inflammatory cytokines listed in Table 1, especially IL-4, IL-6, and TNF-α, also play a major role in pancreatic fibrosis formation. Additional activators and proliferators of PSCs are listed in Table 2 and Table 4.
biomolecules-14-00239-t002_Table 2Table 2An overview of the fibrotic biomarkers involved in the development and progression of chronic pancreatitis. (Symbols in the column “Blood Levels in Patients with CP” correspond to the reference in the column “Biomarkers”; =: no difference; ↑: reduced levels; and ↑: elevated levels).BiomarkersMechanismPancreas-Specific EffectsBlood Levels in Patients with CP CommentMMP-7 [48,87]Enzymes secreted by activated PSCs that degrade the ECM [120].Degradation of basement collagen (type IV) [39,112].= ↑
MMP-9 [39,112,113,114]↑ ↑ = ↑One study found elevated levels in the plasma and not in the serum [114].TIMP-1[15,48,85,87,105,106,113,115,116]Enzymes secreted by activated PSCs that inhibit MMPs [120].Inhibits the proteolytic activity of MMPs. An imbalance between MMP And TIMPs supports the abnormal formation of the ECM [120].↑ ↑ = ↑ = ↑ ↑ =mRNA expression in the pancreas increases with disease progression [121].HA[92,101,117,118]A protein component of the ECM [101].Marker of ECM proliferation.↑ ↑ ↑ =
TGF-β [7,10,24,91,92,94,101,117,122,123]A multipotent growth factor, with various functions, e.g., cell differentiation, proliferation, matrix production, and apoptosis. Promotes the recruitment of inflammatory cells and contributes to fibrosis [124].Activates PSCs leading to fibrosis formation in CP [124].↑ ↑ = ↑ ↑ ↑ ↑ ↑ ↑ =Higher in patients with pancreatic atrophy than in patients with a non-atrophic pancreas [7]. Correlates with the severity of alcoholic CP [91].PDGF[10,12,46,89,94,117,122]A growth factor and mitogen acting on fibroblasts and promoting cell proliferation and migration [125].Acts as a growth factor on PSCs leading to ECM formation and, consequently, fibrosis [101].↑ = ↓ = ↑ ↑ =One paper studied PDGF-AA [122].No correlation between PDGF-BB and alcohol intake [117].MCP-1[7,9,12,24,25,29,48,91,92,101,111]A chemoattractant that recruits an inflammatory infiltrate and initiates inflammation [24,126].Activates PSCs via TNF-β and promotes pancreatic fibrosis [101].= ↓ = = ↑ = = = ↑ ↑ ↑Negative association with alcohol [24]. Treatment with MCP-1 antagonist in rats inhibits pancreatic fibrosis [7].MIC-1[17,106,127,128,129]Part of the TGF-β family. An autocrine regulator of macrophage activation [130].The specific mechanism in the pancreas is not clear [106,129].↑ ↑ ↑ ↑ ↑ =Further elevated in patients with PDAC, making it a potential biomarker [106].M2BP [131,132]A ligand that binds to extracellular proteins such as integrins, collagens, and fibronectin [133].Suggested to be associated with cell-to-cell and cell-to-ECM adhesion and plays a role in the facilitation of fibrosis [134].↑ ↑A novel biomarker of liver fibrosis [135,136].ET-1 [22,137]A mediator with vasoconstrictive and pro-inflammatory properties, secreted by damaged endothelial cells [137].Affects the activation of PSCs and stimulates the migration of PSCs [138].= =Elevated levels seen in smokers [137].EGF[29,89,139]A growth factor that stimulates the proliferation of, e.g., fibroblasts and epithelial cells [140].Regulates both chemoattraction and stimulation of the proliferation of PSCs [141].↑ ↓ ↑
IGF-1[24,48,50,139,142,143,144]A growth factor that plays an important role in many bioactivities such as cell proliferation, differentiation, and survival [145].Stimulates migration and proliferation of PSCs [146].= = = = ↑ = =One study found reduced levels of IGF-1R [144].IGFBP-2 [48,142]Insulin growth factor-binding protein 2↑ ↑
ECM: extracellular matrix; EGF: epidermal growth factor; ET-1: endothelin; HA: hyaluronic acid; IGF: insulin-like growth factor; IGFBP: insulin-like growth factor-binding protein; MCP: monocyte chemoattractant protein; MIC: macrophage inhibitory cytokine; MMP: matrix metalloproteinase; M2BP: mac-2-binding protein; PDAC: pancreatic adenocarcinoma; PDGF: platelet-derived growth factor; TGF: transforming growth factor; and TIMP: tissue inhibitor of metalloproteinases.


### 3.3. Oxidative Stress

Twenty-three studies from 1981 to 2022 examined 34 biomarkers of oxidative stress, of which 23 were studied multiple times. Four articles also included patients with AP [147,148,149,150]. A potential relationship between oxidative stress and pancreatic inflammation has been extensively studied. Research indicates an early occurrence of pancreatic oxidative stress in AP. Free oxygen radicals play a crucial role in regulating the extent of necrosis in acinar cells, the development of pancreatic edema, the sequestration of inflammatory cells within the pancreas, and the release of inflammatory mediators [151]. Additionally, there is growing evidence connecting oxidative stress and CP. The use of antioxidant therapy has been shown to reduce the severity of CP, resulting in less fibrosis in murine models [152], as well as improve the well-being, decrease pain, and improve the overall functioning of patients with CP [153,154]. The biomarkers of oxidative stress are challenging to evaluate, primarily due to their complex metabolism and high turnover, making them difficult to measure in systemic circulation. Most of the biomarkers included in the present review are, therefore, indirect markers of oxidative stress. The low blood antioxidant levels could be attributed to poor nutritional status due to malabsorption, maldigestion, and reduced food intake, often observed in patients with CP. Figure 3 gives a schematic overview of oxidative stress biomarkers associated with CP development.

#### 3.3.1. Lipid Peroxidation

Prolonged exposure to oxygen radicals results in lipid peroxidation and the oxidation of fatty acids in cell membranes. Lipid peroxidation has been the focus of several studies investigating oxidative stress in CP [155]. Many of the biomarkers in Table 3 are primarily byproducts of lipid peroxide; these include TBARS, 4-HNE, MDA, and CD. Fourteen studies have included one of these biomarkers, and, apart from a few non-significant results, all these biomarkers are found to be elevated in patients with CP. Few studies include oxygen radicals in patients with CP. Superoxides, main reactive oxygen species (ROS) in cells, and ROS production in cells after phorbol myristate acetate (PMA) stimulation have all been investigated in patients with CP. These biomarkers are difficult to measure in the blood as they have a very short half-life. Elevated levels were found in all four studies; however, in two of them, the elevation was not significant. On the other hand, PON1 is a free radical-scavenging molecule contributing to the detoxification of free radicals involved in lipid peroxidation [156], and consistently reduced levels of PON1 were found in patients with CP in the studies we analyzed, indicating elevated lipid peroxidation in patients with CP. MDA, superoxide anion, and CAT were also found to be elevated in AP [147,148,149,150]. 

#### 3.3.2. Antioxidation

The reactive superoxide is catalyzed to hydrogen peroxide by superoxide dismutase (SOD). The main ROS scavenger molecule is glutathione (GSH), which is used by glutathione peroxidase (GPX) and catalase (CAT) to reduce/neutralize ROS [157]; see Figure 3. GSH, GPX, CAT, and SOD have been measured in patients with CP in, respectively six, eight, four, and six studies. GSH, GPX, and SOD levels were reduced in patients with CP, while the results on CAT were contradictive, as two studies found no difference, one found lower levels, and one study found elevated levels. The antioxidant capacity in the blood is measured by the ferrin-reducing ability of the plasma (FRAP) and the total peroxyl radical-trapping antioxidant parameter (TRAP). While FRAP is reduced in patients with CP, the TRAP concentrations are no different from the controls. Reduced levels of GSH and elevated levels of CAT and TRAP were found in patients with AP [147,148,149].
biomolecules-14-00239-t003_Table 3Table 3An overview of the oxidative stress biomarkers involved in the development and progression of CP. (Symbols in the column “Blood Levels in Patients with CP” correspond to the reference in the column “Biomarkers”; =: no difference; ↓: reduced levels; and ↑: elevated levels).BiomarkersMechanismBlood Levels in Patients with CPCommentTBARS [122,147,158,159,160,161,162]A byproduct of the lipid peroxidation process [163].↑ ↑ ↑ ↑ ↑ ↑ =TBARS are higher in patients with TCP than in patients with ACP [160].4-HNE [164,165]A byproduct of the lipid peroxidation process [166].↑ ↑Also elevated in RAP, especially during attacks on AP [164].MDA [149,164,165]One of the final products of lipid peroxidation [164].↑ = ↑Elevated levels are also found in pancreatic tissue samples [149].CD [155,157]Primary products in lipid peroxidation in cells [157].= ↑Elevated levels are also found in pancreatic tissue samples [149].ROS [158,167]Reactive oxygen species= ↑Difficult to measure in the blood due to a short half-life.O2−[148,167]Reactive oxygen species molecule [148].↑ ↑Elevated in both PMA-stimulated and resting neutrophils [167].GSH [157,159,160,161,168]The main ROS scavenger. Used by GPX to metabolize H_2_O_2_ and lipid hydroperoxides to water/alcohols [157].= ↓ ↓ ↓ ↓
GPX [157,159,160,162,164,167,169,170]Catalyzes hydrogen peroxide to oxygen and water and, therefore, has an important function in the protection against oxidative stress [167].↓ ↓ ↓ ↓ ↓ ↓ = ↓
CAT [150,157,167,169]= = ↑ ↓
SOD [157,159,160,161,164,167,169]Catalyzes the dismutation of superoxide anions to hydrogen peroxide [157].= ↓ ↓ ↓ = ↑ =One study found elevated serum SOD and lower levels of erythrocyte SOD in patients with CP [161].PON1 [156,157]An HDL-associated enzyme. Plays a role in the hydrolyzation of active oxidized phospholipids and in the destruction of lipid hydroperoxides and H_2_O_2_ and prevents oxidation of LDL [156].↓ ↓
TRAP [147,165]Total peroxyl radical-trapping antioxidant parameter.= =
FRAP [122,158,161,164]Ferrin-reducing ability of the plasma. A measurement of the non-enzymatic antioxidant capacity of the plasma [158].↓ ↓ ↓ ↓Lower levels are also observed in patients with RAP [164].Vitamin A [42,161,162,169,171]Blood antioxidant↓ ↓ ↓ ↓ ↓Dietary-dependentVitamin C [147,159,160,161,164,169]Blood antioxidant↓ ↓ ↓ = = =Dietary-dependentVitamin E [42,161,162,169,170,171]Blood antioxidant↓ ↓ ↓ ↓ ↓ ↓Dietary-dependentZink [162,169,171]Blood antioxidant= = =Elevated levels in patients with RAP [171].Copper [162,169,171]Induces oxidative stress by increasing ROS [172].↑ ↑ =Reduced levels in patients with RAP [171].Selenium [162,169,171]Blood antioxidant↓ ↓ ↓
Homocysteine [158,168]Amino acid mediator in the synthesis of GSH.= ↑
Cysteine [158,168]Essential amino acid necessary for the formation of GSH.↓ ↓
Methionine [49,168,173]Essential amino acid necessary for the formation of GSH.= ↓ ↓One study only found elevated levels in TCP [168].β-carotene [156,169,171]Blood antioxidant↓ ↓ ↓
ACP: alcoholic chronic pancreatitis; CAT: catalase; CD: conjugated dienes; FRAP: ferrin-reducing ability of the plasma; GPX: glutathione peroxidase; GSH: glutathione; MDA: malondialdehyde; $O_{2}^{-}$: ion oxide; PON1: paraoxonase; PMA: phorbol myristate acetate; RAP: recurrent acute pancreatitis; ROS: reactive oxygen species; SOD: superoxide dismutase; TBARS: Thiobarbituric acid-reactive substances; TCP: tropical chronic pancreatitis; TRAP: total peroxyl radical-trapping antioxidant parameter; and 4-HNE: 4-hydroxynonenal.
biomolecules-14-00239-t004_Table 4Table 4An overview of the biomarkers examined once.InflammationFibrosisOxidative stressGroupBiomarkerExpression Changes in CPGroupBiomarkerExpression Changes in CPGroupBiomarkerExpression Changes in CPGroupBiomarkerExpression Changes in CPInterleukinsIL-5 [9]nsAdhesionmoleculesCD44 [62]↓Componentsof the ECMCollagen IV [87]↑AntioxidantsGR [157]nsIL-7 [9]nse-selectin [22]nsFibronectin [87]↑Xanthine [171]↓IL-13 [9]nsVCAM [22]nsLaminin [117]↑Β-cyproxanthine [171]↓IL-15 [29]↓Complement factorsC1q [9]↓MMP-1 [112]↑Lycopene [171]↓IL-16 [9]nsC3 [9]nsMMP-2 [47]↑SH groups [147]↓IL-18 [13]↑C4 [9]↑MMP-3 [112]nsLipidperoxidationOx-LDL/LDL [157]↑IL-23 [75]↑C4BPA [174]↑PICP [87]nsROOH [170]↑CytokinesTNF-β [9]↓C5 [9]↑PINP [87]nsLipid peroxide [170]↑GCSF [78]nspro-C3 [118]nsTHBS1 [85]nsProteindamage3-NT [157]↑MCSF [78]nsPro-C5 [175]nsTPS [18]↑Carbonyls [158]↑IFN-α [29]nsProperdin [9]↓TSP-2 [87]↑OthersNitrites [165]↑ChemokinesCCL5 [85]nsAdipokinesOmentin [95]↑OthersAZGP1 [85]↑


CXCL16 [176]nsOthersANG-1 [89]↓CCN1 [87]ns


IP10 [12]nsHMGB1 [10]nsCCN2 [87]↑


MCP-3 [9]nsLBP [85]nsPLG [87]ns


MIP-1β [12]nsLTF [85]↑





MIP-3α [11]↑RORγT [75]↑





PPBP [85]nsSTAT3 [75]↑





RBP-4 [143]nsYKL-40 [26]ns





GrowthfactorsIGF-2 [48]nsCD40L [29]↑





IGFBP1,3 [48]↑








ns: non-significant; ANG-1: angiopoietin; AZGP: zinc-α2-glycoprotein 1; CCL: C-C motif chemokine ligand; CNN: cellular communication network; CXCL: C-X-C motif ligand; G-CSF: granulocyte colony-stimulating factor; GR: glutathione reductase; HGMB: high-mobility group box; IGF: insulin-like growth factor; IGFBP: insulin-like growth factor-binding protein; IL: interleukin; INF: interferon; IP: induced protein; LBP: lipopolysaccharide-binding protein; LTF: lactoferrin; MCP: monocyte chemotactic protein; MCSF: macrophage colony-stimulating factor; MIP: macrophage inflammatory protein; MMP: matrix metalloproteinase; PLG: plasminogen; PICP/PICP: procollagen type I C-terminal propeptide/N-terminal propeptide; PPBP: pro-platelet basic protein; RBP: retinol-binding protein; ROOH: lipid hydroperoxide; RORγ: retinoic acid receptor-related orphan receptor gamma; SH groups: thiols; STAT: signal transducer and activator of transcription; THBS: thrombospondin; TNF: tumor necrosis factor; TSP-2: tissue polypeptide specific antigen; TSP: thrombospondin; VCAM: vascular cell adhesion protein; and 3-NT: 3-nitrotyrosine.


## 4. Discussion

The identification of a biomarker for the early diagnosis of CP is of great importance, as it has the potential to make a significant impact on disease prevention and intervention. Moreover, an early-stage CP biomarker could facilitate clinical trials with anti-fibrotic, anti-inflammatory, and antioxidative therapies. Several biomarkers of inflammation, fibrosis, and oxidative stress have been studied so far to improve our understanding of the transition from AP to CP. Herein, we present a general overview of established inflammatory, fibrotic, and oxidative stress biomarkers associated with the progression to CP as well as a brief presentation of the most promising biomarkers. It is important to note that 50% of patients with CP have not had an episode of AP prior to their diagnosis; therefore, some of the biomarkers associated with CP might not be associated with AP [3].

Studies on CP biomarkers present varying results. Some inflammatory biomarkers have been linked to CP development, mainly IL-6, IL-8, IL-10, IL-12, TNF-α, ICAM, fractalkine, and some adipokines. All the findings are not coherent; therefore, we found no conclusive inflammatory pattern of CP. Several biomarkers of fibrosis are used to determine fibrotic development in patients with CP. The main biomarkers of circulating ECM components are MMPs, TIMP-1, and HA, and the main stimulators of PSCs are TGF-β, PDGF-BB, and MCP-1. In addition, oxidative stress is increased, and antioxidant capacity is lowered in CP. The exact impact of oxidative stress on disease progression and development is still not thoroughly understood.

CP is histologically characterized by the loss of acinar cells, irregular interlobular fibrosis, infiltration of inflammatory cells, and, eventually, also by the loss of intralobular ducts and islets [177]. The pattern of fibrosis varies depending on the type of cell primarily affected by the injury. Different causes of CP result in different fibrotic patterns. For instance, alcoholic CP is more likely to exhibit inter(peri)lobular fibrosis, whereas nonalcoholic CP more frequently shows periductal or intralobular fibrosis [119]. After exposure to risk factors, such as smoking, alcohol, intraductal obstruction, or other injuries, oxidative stress occurs, leading to the necrosis and apoptosis of pancreatic cells. This triggers the activation and infiltration of inflammatory cells. Once the inflammatory cells are activated, they secrete cytokines, growth factors, and other molecules that promote the differentiation and activation of the PSCs. Activated PSCs excessively secrete and synthesize extracellular matrix (ECM). PSCs cause ECM to be continuously remodeled, resulting in fibrosis. Activated PSCs secrete matrix metalloproteinases (MMPs), enzymes which degrade the ECM, and tissue inhibitors of MMPs (TIMPs), enzymes which inhibit MMPs. An imbalance between MMPs and TIMPs can lead to the abnormal formation of the ECM [119]. Oxidative stress occurs when ROS damage cell lipids and proteins. Lipid peroxidation is the process by which ROS interact with polyunsaturated fatty acids. Lipid peroxidation in cell membranes causes cell damage [158]. Antioxidants neutralize ROS molecules, having a protective effect on CP. The processes of fibro-inflammation and oxidative stress are intertwined, as oxidative stress causes inflammation but also directly activates PSCs [178], while inflammation causes the formation of ROS and activates PSCs; see Figure 4.

This review presents promising CP biomarkers. The three most promising biomarkers of inflammation seem to be IL-6, IL-8, and fractalkine, meanwhile, those for fibrosis are TGF-β1, HA, and MIC-1 and, for oxidative stress, TBARS, FRAP, and GPX. However, these candidates require further investigation before they can be implemented in clinical practice.

Several methodological limitations must be taken into consideration when interpreting the findings of current research efforts. The studies included in this review have marked variability in their design, population, and CP diagnosis criteria. The lack of standardization and proper characterization of patients included in studies can lead to biased results and inaccurate conclusions. Disease stage standardization is particularly crucial, as biomarker patterns may differ depending on whether samples were collected in the early or later stages of the disease or during or between flare-ups. Due to the fact that CP can be difficult to identify in its early stage, most clinical studies included patients with advanced CP or a flare-up, as these patients attend the sites of research for treatment. One study found elevated levels of TGF-β1 in only mild and moderate CP, the assumed reason for this is the replacement of pancreatic tissue with fibrotic tissue. Advanced CP is characterized by fibrosis formation, which might play a role in the levels of biomarkers, as less pancreatic tissue is present to produce cytokines and other factors. Disease severity should, therefore, be a target for standardization and should at least be characterized in participating patients.

How biomarkers are measured can also affect the accuracy of the outcome. Measuring biomarkers can be challenging at times, especially when certain markers have a short half-life and are difficult to measure correctly in the plasma or the serum. This can lead to inaccurate results. The studies included in this review have used different methods, which might explain the varying results which the same biomarker has in different studies.

Additionally, CP is a complex disease with different etiologies that may have different sets of biomarker patterns, and factors such as smoking, alcohol, diabetes, age, gender, and BMI are not matched in all the studies, which may lead to confounding effects, as inflammation and oxidative stress may be present due to other causes. It is also important to note that the interaction between cytokines and their specific inhibitors in the plasma makes the accurate analysis and interpretation of cytokine levels and activity difficult.

Many cytokines and MMPs primarily exert paracrine effects, which are not necessarily expressed in the systemic circulation. Oxidative stress is typically localized and organ-specific. By implementing the notion that the levels of certain molecules in the bloodstream may not accurately reflect the specific processes occurring in the organs, it becomes clear that many of the biomarkers included in this review may also be elevated in other diseases and conditions. Therefore, it is important to consider factors such as smoking, alcohol use, and diabetes when comparing biomarker levels to those of healthy individuals, as these typically are present in some form in patients with CP.

The development of CP can be associated with disturbances in the flow of pancreatic fluid from the acinar cells through the side branches and ducts in the pancreas. The disturbances can be caused by strictures, the formation of intraductal stones, or intraductal hypertension caused by (extra)pancreatic collections. Pancreatic obstruction is linked to enhanced levels of inflammatory, fibrogenic, and oxidative stress markers in the pancreas. Interventional procedures like endoscopic retrograde cholangiopancreatography or extracorporeal shock-wave lithotripsy are often used to reduce pancreatic duct obstruction, thereby reducing intraductal pressure and, presumably, decreasing the levels of the measured biomarkers. Their effect on biochemical patterns in CP has not been extensively studied and remains a topic of interest.

Some of the studies included in the present review also examined biomarkers in patients with AP and RAP. These studies indicate that some of the biomarkers elevated in CP are also elevated in patients with AP (e.g., IL-6, IL-2, MCP-1) and in patients with RAP (e.g., MCP-1, 4-HNE, MDA). This overlap in biomarkers makes it challenging to use them as early predictors of CP. Further prospective evaluation of a large and well-characterized cohort of patients with AP, RAP, and CP may clarify the differences in the expression levels of biomarkers within these separate entities.

The current literature review mainly includes cross-sectional studies on patients with advanced CP, which limits the ability of this work to accurately assess disease progression. Future longitudinal studies including patients with AP, RAP, and early CP are essential in order to provide valuable insights into the time-dependent interplay between inflammatory, fibrogenic, and oxidative stress biomarkers and the development of CP. Such a prospective and prolonged follow-up may give us valuable insight and a rationale for interventional studies [179,180].

CP is a rather heterogenous disease, and, within this diagnosis, several phenotypes as well as clinical challenges exist. Future research should be aimed at associating different phenotypes (obstructive, inflammatory, painful, etc.) with different biomarker profiles.

## 5. Conclusions

The fibro-inflammatory process involved in the progression from acute to recurrent acute and, ultimately, chronic pancreatitis is epidemiologically and clinically evident. Several biomarkers have been investigated in an attempt to unravel this process, but their exact role in the continuous process of inflammation, fibrosis, and oxidative stress towards CP is not yet fully established. Currently, there is no reliable biomarker specifically indicative of chronic pancreatitis. The development of chronic pancreatitis remains poorly understood, and a better understanding of it may help researchers identify new targets for intervention.

## Figures and Tables

**Figure 1 biomolecules-14-00239-f001:**
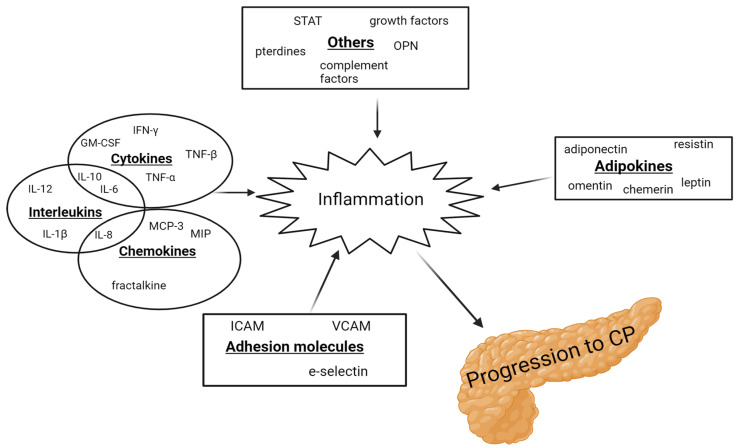
Schematic illustration of the inflammatory biomarkers involved in the progression to CP. GM-CSF: granulocyte-macrophage colony-stimulating factor; ICAM: intracellular adhesion molecule; IFN: interferon; IL: interleukins; MCP: monocyte chemotactic protein; MIP: macrophage inflammatory protein; OPN: osteopontin; STAT: signal transducer and activator of transcription; TNF: tumor necrosis factor; and VCAM: vascular cell adhesion molecule.

**Figure 2 biomolecules-14-00239-f002:**
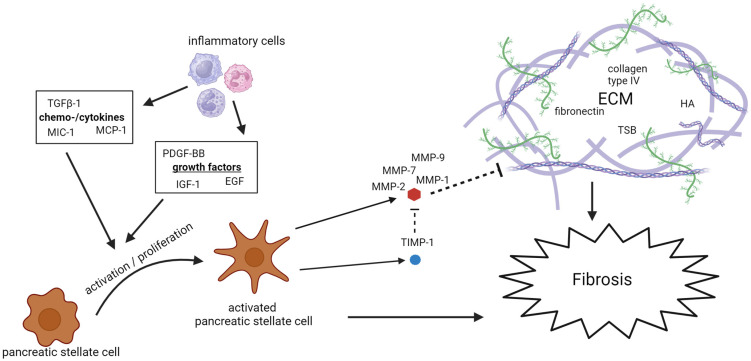
Schematic overview of the fibrotic biomarkers associated with the development of CP. ECM: extracellular matrix; EGF: epidermal growth factor; HA: hyaluronic acid; IGF: insulin-like growth factor; MCP: monocyte chemotactic protein; MIC: macrophage inhibitory cytokine; MMP: matrix metalloproteinase; PDGF: platelet-derived growth factor; TGF: tumor growth factor; TIMP: tissue inhibitors of metalloproteinases; and TSP: tissue polypeptide specific antigen.

**Figure 3 biomolecules-14-00239-f003:**
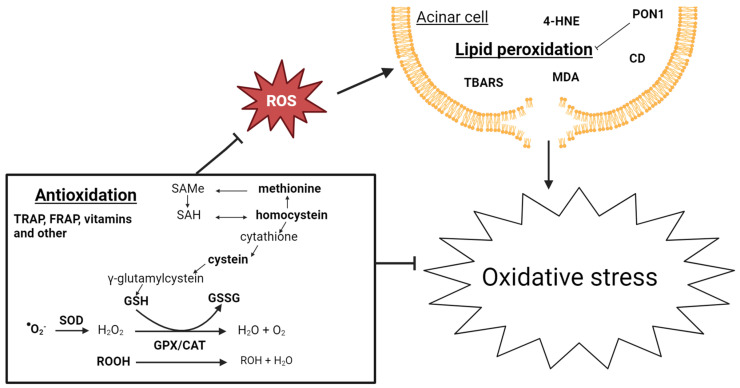
Schematic overview of biomarkers of oxidative stress associated with CP development. The biomarkers included in this review are marked with a bold font. CAT: catalase; GPX: glutathione peroxidase; GSH: glutathione; GSSG: glutathione disulfide; ROOH: hydroperoxides; SAH: S-adenosyl homocysteine; SAMe: S-adenyl methionine; and SOD: superoxide dismutase.

**Figure 4 biomolecules-14-00239-f004:**
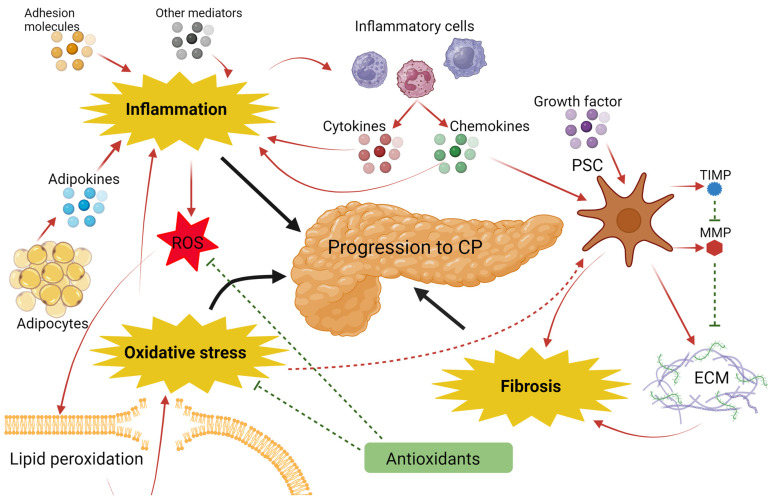
Schematic overview of the fibro-inflammatory and oxidative stress process in the development of CP. Created with BioRender.com. ECM: extracellular matrix; MMP: matrix metalloproteinase; PSC: pancreatic stellate cells; ROS: reactive oxygen species; and TIMP: tissue inhibitor of metalloproteinase.

## Data Availability

No new data were created or analyzed in this study. Data sharing is not applicable to this article.

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
