# Peer review of "Circulating Biomarkers Involved in the Development of and Progression to Chronic Pancreatitis—A Literature Review"

_biomolecules, 2024, doi:10.3390/biom14020239_

Round 1
Reviewer 1 Report
Comments and Suggestions for Authors
This is a great work which summarized the potential biomarkers that can either predict the development of CP or the severity of pancreatic injury. CP is not as common as AP, the incidence is low and the mechanisms of initiation and progression have not been well-understood. I have few comments and just suggest to be accepted after the revision.
1.What’s the main purpose of picking up “oxidative stress” as one of the keywords in this study?
2.The author could add a few discuss on the relations between those biomarkers and their roles in CP therapy. Since many cytokines can be used for the detection/evaluation, whether some specific markers could be considered as interventional targets.
3.Are there any ways to clarify the difference/different expression levels of those biomarkers within AP, RAP and CP? The closed links between them, especially RAP and CP may cause the similarity of systemic changes. The separation will be helpful.
4.Any clues between circulating biomarkers with clinical symptoms in CP patients, such as pain, pancreatic exocrine/endocrine insufficiency?
Reviewer 2 Report
Comments and Suggestions for Authors
This is an interesting review of circulating biomarkers in the development and progression of CP.
The authors have compiled a nice and very informative review article. The conceptualization, presentation, and language are all appreciable. The authors have focused on the biomarkers of inflammation, fibrosis, and oxidative stress in patients with CP, and have done justice to the collection of information scanned for almost four decades.
The results are presented very clearly and explicitly, highlighting the important biomarkers like IL-6, TNF-A, and Leptin in case of inflammation among others. Additional information in the form of tables and well-presented figures reflects the in-depth observations carried out.
The discussion section is also well-written highlighting the importance and limitations of biomarker identification.
Although there is no certain biomarker for CP due to the limited understanding of the etiology of the disease and there is a long way to go for conclusive findings for biomarkers for different stages or causes of CP, reviews like this will help pave the way for future research.
I recommend acceptance of the article in its current form.
Author Response
Dear editor/reviewers,
We would like to thank you for your thoughtful and positive feedback on our review article. We appreciate your acknowledgement of the clarity in conceptualization, presentation and language. Your recommendation for the acceptance of the article in its current form is greatly appreciated
Sincerly,
Valborg Vang Poulsen
Reviewer 3 Report
Comments and Suggestions for Authors
The literature review article "Circulating biomarkers involved in the development of- and progression to chronic pancreatitis- a literature review" aims to comprehensively provide an overview of circulating molecules that have been investigated as potential biomarkers for the transition of acute pancreatitis to chronic pancreatitis or as biomarkers for chronic pancreatitis alone. The article is well written and the authors provide an adequate overview of the current knowledge in the field. However, minor revisions would enhance the manuscript prior to publication. My comments are below.
- The first sentence of the abstract is slightly confusing after reading it for the first time. Consider revising (eg., CP is the end stage of continuous inflammation and fibrosis in the pancreas evolving from acute, to over recurrent acute, to early stage CP...)
- Lines 34-36. It is unclear if RAP happens in 21% of patients with AP and CP happens in 36% of patients with AP or if 36% of the patients with RAP go on to be diagnosed with CP. Also, when stating 21% of patients, please clarify what defines patients. Are these all patients diagnosed with AP?
- Lines 36-38. Similar to Lines 34-36, does the 16% and 50% refer to the 21% of patients that have RAP? If so, the current wording seems to dilute the patient pool that actually go on to be diagnosed with CP after being diagnosed originally with AP. Perhaps using actual patient numbers in the U.S. or worldwide (ie., 30 million patients, 100 million patients, etc...) would enhance the impact. Please consider clarifying/revising.
- Line 71. The acronym "VEGF" is spelled incorrectly.
- Figure 1. While the authors list the interleukins in the Results section, they are not listed in the Interleukins diagram within Figure 1. Also, "Interleukins" text is a different format than "Adipokines, Others, Cytokines" in Figure 1. Please consider revising.
- Table 1. The acronym "VEGF" is spelled incorrectly.
- Line 127. Consider elaborating on which 13 molecules were studied multiple times. Are the majority of these cytokines, growth factors, adipokines? Consider emphasizing the top 3-5 molecules of the 13 that were studied the most.
- The addition of a single statement to the Methods indicating that all figures were generated using BioRender and the removal of this statement from the figure legends would enhance the manuscript.
- Line 144. Consider adding which MMPs were studied since the authors state that MMP3 was the only MMP found to be decreased. It makes the reader wonder which MMPs were elevated.
- Line 151. the authors do not state in what disease elevated components of the ECM were found. The text assumes it to be CP, but since the authors also discuss control groups and AP it should be clarified which group elevated ECM components were found.
- Line 154. the authors state "pancreatitis" but do not clarify if this is AP, RAP, or CP. Since the manuscript discusses three types of pancreatitis, clarification here of which pancreatitis the authors are referring to is needed.
- Line 159-160. Please clarify which cells in the pancreas are secreting MMP and TIMP in this context. The authors also list other molecules that are secreted or found in the pancreas to be related to ECM remodeling. Since there are various cell types in the pancreas capable of secreting a number of the molecules listed, clarification here of which cell types are secreting which molecules would enhance the manuscript.
- Line 198 has an unusual indent.
- Figure 3. It is unclear what is being depicted in the image labeled "Lipid Peroxidation". If this is an acinar cell please label, or clarify which cell this is referencing.
- Line 232. "figure 3" should be "Figure 3" to be consistent with the rest of the text references to figures.
- Consider adding a statement or discussing biomarkers that may be common in both AP and CP and therefore warrant further investigation as early predictors to CP. Or, if these have already been studied consider adding a statement pertaining to this and therefore adding to the increased difficulty and importance to identify an early predictor.
- Line 282. Do these stressors affect a specific cell type in the pancreas more than others, or are specific stressors specific to cell type? If so, consider adding a statement discussing this. For example, it could be interesting to know if ROS affects endocrine cells more than stellate cells and therefore leads to increased metabolic disease versus fibrosis, etc...
Comments on the Quality of English Language
The quality of the English is good. Minor edits to the text are stated in my comments.
Reviewer 4 Report
Comments and Suggestions for Authors
"Circulating biomarkers involved in the development of- and progression to chronic pancreatitis – a literature review" is a very well written manuscript providing a comprehensive overview on potential circulating factors for the development and progression of chronic pancreatitis. The Authors provided a consistent background and analysed several biomarkers potentially involved in the process, describing the pathogenic role of each factor discussed. Figures and tables are well done and easily readable.
Author Response
Dear Reviewer,
Thank you for your positive feedback on our manuscript. We appreciate your acknowledgement of the comprehensive overview, consistent background, and thorough analysis of potential circulating factors. Your comments on the quality of the figures and tables are encouraging. We are grateful for your valuable insights
Best regards
Valborg Vang Poulsen